# Exploring the Role of *Debaryomyces hansenii* as Biofertilizer in Iron-Deficient Environments to Enhance Plant Nutrition and Crop Production Sustainability

**DOI:** 10.3390/ijms25115729

**Published:** 2024-05-24

**Authors:** Jesús Sevillano-Caño, María José García, Clara Córdoba-Galván, Carmen Luque-Cruz, Carlos Agustí-Brisach, Carlos Lucena, José Ramos, Rafael Pérez-Vicente, Francisco Javier Romera

**Affiliations:** 1Departamento de Agronomía (DAUCO) María de Maeztu Unit of Excellence 2021–2024, Campus de Excelencia Internacional Agroalimentario de Rabanales (ceiA3), Universidad de Córdoba, 14071 Córdoba, Spain; o42secaj@uco.es (J.S.-C.); b82cogac@uco.es (C.C.-G.); b72lucrc@uco.es (C.L.-C.); cagusti@uco.es (C.A.-B.); b42lulec@uco.es (C.L.); ag1roruf@uco.es (F.J.R.); 2Departamento de Química Agrícola, Edafología y Microbiología, Campus de Excelencia Internacional Agroalimentario de Rabanales (ceiA3), Universidad de Córdoba, 14071 Córdoba, Spain; mi1raruj@uco.es; 3Departamento de Botánica, Ecología y Fisiología Vegetal, Campus de Excelencia Internacional Agroalimentario de Rabanales (ceiA3), Universidad de Córdoba, 14071 Córdoba, Spain; bv1pevir@uco.es

**Keywords:** yeast, biofertilization, Fe deficiency, ISR, plant growth promotion, plant biostimulation

## Abstract

The European “Green Deal” policies are shifting toward more sustainable and environmentally conscious agricultural practices, reducing the use of chemical fertilizer and pesticides. This implies exploring alternative strategies. One promising alternative to improve plant nutrition and reinforce plant defenses is the use of beneficial microorganisms in the rhizosphere, such as “Plant-growth-promoting rhizobacteria and fungi”. Despite the great abundance of iron (Fe) in the Earth’s crust, its poor solubility in calcareous soil makes Fe deficiency a major agricultural issue worldwide. Among plant promoting microorganisms, the yeast *Debaryomyces hansenii* has been very recently incorporated, for its ability to induce morphological and physiological key responses to Fe deficiency in plants, under hydroponic culture conditions. The present work takes it a step further and explores the potential of *D. hansenii* to improve plant nutrition and stimulate growth in cucumber plants grown in calcareous soil, where ferric chlorosis is common. Additionally, the study examines *D. hansenii*’s ability to induce systemic resistance (ISR) through a comparative relative expression study by qRT-PCR of ethylene (ET) biosynthesis (*ACO1*), or ET signaling (*EIN2* and *EIN3*), and salicylic acid (SA) biosynthesis (*PAL*)-related genes. The results mark a significant milestone since *D. hansenii* not only enhances nutrient uptake and stimulates plant growth and flower development but could also amplify induced systemic resistance (ISR). Although there is still much work ahead, these findings make *D. hansenii* a promising candidate to be used for sustainable and environmentally friendly integrated crop management.

## 1. Introduction

Crop production faces a formidable challenge: The need to satisfy the escalating global demand for food. Historically, this goal has been mainly achieved by using chemical fertilizers, frequently in an abusive way. However, in alignment with the European “Green Deal”, contemporary policies are gravitating toward fostering a more sustainable and environmentally conscious agricultural approach. This entails a substantial reduction in the utilization of chemical fertilizers. Therefore, the pursuit of alternative strategies becomes imperative.

Among these alternative strategies is the optimization of the rhizosphere [1,2], since it is a concept substantiated by the research conducted on harnessing microorganisms to enhance crop sustainability. Beneficial rhizosphere microorganisms play a pivotal role in improving plant nutrition and fortifying plants against both pathogens and herbivores [1,2,3,4,5,6,7,8,9,10,11].

Plant-growth-promoting (PGP) rhizobacteria (PGPR) and fungi (PGPF) emerge as essential contributors to plant growth improvement and adaptation to adverse conditions [1]. The significance of these PGP microorganisms in enhancing plant growth predominantly arises from their ability to boost nutrient uptake and induce systemic resistance, a phenomenon known as “Induced Systemic Resistance” (ISR) [10,12,13]. Besides ISR, plants can also induce other systemic resistance responses to fight against pathogen and herbivorous attacks, known as “Systemic Acquired Resistance” (SAR) and “Systemic Wound Resistance” (SWR), respectively. Typically, ISR is mediated by ethylene (ET) and jasmonic acid (JA); SAR by salicylic acid (SA); and SWR by JA [1,14,15].

Among the PGP microorganisms are mycorrhizal fungi and *Rhizobium* spp., which, through their symbiosis with plant roots, significantly enhance phosphorus (P) and nitrogen (N) nutrition, respectively [16,17]. Beyond these, a diverse array of free-living mutualistic microbes effectively enhances plant nutrition through various mechanisms, which include the release of nutrient-solubilizing compounds and the modification of root architecture and physiology [1,11,18,19,20,21,22,23].

Relative to the different PGPRs, extensive research has delved into the role of *Bacillus* spp., *Klebsiella* spp., and *Pseudomonas* spp. as pivotal players in promoting plant health and serving as biological control agents (BCAs). These genera have been scrutinized for their ability to induce ISR in a wide range of crops [24,25,26,27,28,29,30,31,32]. In the context of PGPF, particular attention has been directed toward mycorrhiza [33,34,35] and *Trichoderma* [36,37,38,39,40,41].

Among the nutrient disorders in plants, iron (Fe) deficiency represents a prominent agricultural issue all over the world [42,43]. Despite being the fourth most abundant element in the Earth’s crust, Fe primarily exists in its oxidized form as Fe^3+^, with poor solubility under the basic pH conditions commonly found in calcareous soils [44]. Plant classification, based on their strategies for Fe acquisition from the soil, has divided them into two categories: Strategy I, which is employed by all higher plants except grasses, and Strategy II, exclusive to grasses [45]. Strategy I is characterized by the necessity to convert the prevalent Fe^3+^ form in soil to Fe^2+^ prior to its absorption. This conversion is facilitated by a ferric reductase located in the plasma membrane of epidermal root cells. In Arabidopsis, this reductase is encoded by *AtFRO2* [46,47]. Once Fe^3+^ has undergone reduction, it is transported into the cells through a Fe^2+^ transporter, which is encoded by *AtIRT1* in Arabidopsis [46]. Graminaceous plants, such as rice and barley, employ a distinctive Fe uptake mechanism involving the synthesis of high-affinity Fe^3+^ chelators known as mugineic acid family phytosiderophores (MAs). These MAs serve to solubilize Fe^3+^ in the rhizosphere, resulting in the formation of Fe^3+^-MA complexes that are subsequently absorbed by the plant’s roots [45].

To cope with Fe deficiency, Strategy I plants develop both morphological and physiological responses in their roots, aimed at facilitating their mobilization and acquisition [45,48]. These responses are regulated by hormones and signaling molecules, mainly ethylene (ET), auxin, and nitric oxide (NO), that exert their function through bHLH transcription factors such as FIT (**F**ER-LIKE **I**RON DEFICIENCY INDUCED **T**RANSCRIPTION FACTOR), which form heterodimers with bHLH38, 39, 100, and 101 [46,48,49,50,51]. Besides these transcription factors, MYB72, dependent on FIT [52], also plays a role in the Fe deficiency response by regulating coumarin synthesis, which acts as chelating and reducing Fe agents, improving its solubility for plants [53,54,55].

Fe deficiency responses and ISR share common regulators such as ET, auxin, NO [1,56,57,58,59], and the transcription factor *MYB72* [1,22,26,48,60]. This makes possible a cross-talk by which the Fe acquisition genes can be induced by both Fe deficiency and exposition to ISR-eliciting microbes [7,8,11,20,22,61,62].

Over the last few years, a diverse array of fungal strains has been incorporated into the PGPF list. In this way, the non-pathogenic strain *Fusarium oxysporum* FO12 has not only been characterized as a potential BCA against Verticillium wilt olive, a fungal vascular disease caused by the soilborne pathogen *Verticillium dahliae* [63,64,65], but also by its capacity to enhance growth and trigger Fe deficiency responses in cucumber (Strategy I) and rice (Strategy II) plants [7,8]. In addition, *Metarhizium brunneum* strain EAMa 01/58-Su, a recognized BCA for its effectiveness to infect a wide range of arthropod pests by contact [66,67,68], has garnered attention for its role as PGPF and for its multifunctionality, inducing Fe deficiency responses and providing protection against pests in plants by priming [10,11].

Recently, Lucena et al. [2] have identified two yeast species as potential biofertilizer candidates. These authors evaluated the effectiveness of three yeast species, *Hansenula polymorpha*, *Debaryomyces hansenii,* and *Saccharomyces cerevisiae*, in inducing significant physiological and morphological responses to Fe deficiency. *Debaryomyces. hansenii* and *H. polymorpha* showed a great capacity to trigger several Fe deficiency responses in cucumber (*Cucumis sativus* L. cv. Ashley) plants, such as ferric reductase activity and rhizosphere acidification, and to increase the expression of Fe acquisition genes, including *FRO1* (ferric reductase oxidase 1), *IRT1* (iron-regulated transporter 1), and *HA1* (H^+^-ATPase 1). Furthermore, both *D. hansenii* and *H. polymorpha* fostered the development of subapical root hairs, a characteristic morphological response to Fe deficiency [2] (Figure 1).

In this work, we take a further step towards optimizing the use of yeasts as growth promoters and for enhancing plant nutrition. Although, both *D. hansenii* and *H. polymorpha* showed a remarkable ability to induce Fe deficiency responses, this study specifically explores the potential of *D. hansenii* to enhance plant nutrition and stimulate plant growth in cucumber plants grown in calcareous soil, where ferric chlorosis is frequent. This approach is substantiated by *D. hansenii’s* exceptional tolerance to various abiotic stress factors and its prevalence in saline soils. Besides this, we have explored the ability of this yeast to elicit ISR. For this, we have conducted a comparative expression study of several genes related to ET synthesis (*ACO1*) and signaling (*EIN2* and *EIN3),* as well as a SA synthesis-related gene, *PAL.* SA is directly related to the induction of the SAR typically induced in response to pathogen attack [1]. The results presented in this work show for the first time the ability of a yeast, *D. hansenii,* not only to improve nutrient acquisition but also to boot ISR in cucumber plants. These findings convert *D. hansenii* as an ideal candidate for its inclusion in integrated crop management, promoting sustainable and environmentally friendly agricultural practices.

## 2. Results

Previous results from our group highlighted the ability of the yeast *D. hansenii* to induce Fe deficiency responses [2]. Nevertheless, its ability to compete in the rhizosphere with other coexisting microorganisms and to promote growth in these conditions remained unexplored until the culmination of this study. The validation of this yeast’s competence in the rhizosphere was a pivotal milestone towards its prospective application as an Fe biofertilizer, thus serving as the primary focus of this research.

### 2.1. Debaryomices hansenii Effect on Plant Growth Promotion, Flowers Development and Chlorophyll Content

In Figure 2, the fresh and dry weight data of the aerial part at 20 and 28 d after the 1st inoculation are depicted. At 20 d after the 1st inoculation, there were significant differences in the fresh weight of cucumber plants inoculated with *D. hansenii* compared to their respective controls, both in sterile soil (F_1,7_ = 62.38, *p* ≤ 0.001) and in non-sterile soil (F_1,9_ = 67.35, *p* ≤ 0.001) (Figure 2A). In general, the results obtained at 28 d after the 1st inoculation were similar to those observed at 20 d, with greater growth observed in plants inoculated in sterile soil once (F_1,8_ = 2.97, *p* = 0.1284) and twice (F_1,7_ = 6.03, *p* ≤ 0.05), as well as in non-sterile soil with one (F_1,9_ = 45.18, *p* ≤ 0.001) and two inoculations (F_1,9_ = 45.92, *p* ≤ 0.001) (Figure 2B).

While control plants cultivated in non-sterile calcareous soil displayed diminished growth compared to those grown in sterile soil, it is noteworthy that the growth of plants inoculated in sterile and non-sterile soil was remarkably similar in terms of dry weight. This suggests that soil inoculation with *D. hansenii* effectively mitigates the growth delay observed in control plants grown in non-sterile soil.

The dry weight data of the aerial part (Figure 2C,D) reflect a consistent positive growth trend in inoculated plants compared to their non-inoculated controls, similar to what was observed in the fresh weight graphs (Figure 2A,B). This trend is further supported by the statistical results from Fisher’s LSD test for plants harvested at 20 d after the 1st inoculation (Figure 2C,D) in both sterile soil (F_1,5_ = 20,826.92, *p* ≤ 0.001) and non-sterile soil (F_1,5_ = 97.84, *p* ≤ 0.001).

The same positive trend in dry weight is also evident at 28 d after the 1st inoculation (Figure 2D) in sterile soil with one (F_1,5_ = 198.49, *p* ≤ 0.001) and two inoculations (F_1,5_ = 27.02, *p* ≤ 0.01). Furthermore, significant differences were observed in non-sterile soil with one (F_1,5_ = 30.83, *p* ≤ 0.01) and two inoculations (F_1,5_ = 31.16, *p* ≤ 0.01).

In addition to the previously measured growth parameters, we conducted a final time count of the number of flowers developed in each treatment (Figure 3). The impact of the yeast is clearly visible when comparing the inoculated plants to the control group. Notably, a significantly higher number of flowers was observed in *D. hansenii* inoculated plants in comparison with their respective controls in both sterile and non-sterile soil (F_1,9_ = 32.00, *p* ≤ 0.001).

Finally, leaf chlorophyll content was measured over the 28-day course of the experiment by using an SPAD-502 Plus meter (Minolta, Osaka, Japan). Figure 4 provides a visual representation of the evolution of chlorophyll content over time. Notably, it becomes apparent that both the inoculated plants grown in sterile and non-sterile soil consistently exhibit higher SPAD values in comparison to their respective control groups throughout the entire experiment. The highest chlorophyll content was obtained in those inoculated plants cultivated in sterile soil.

As shown in Figure 4, higher differences in SPAD between inoculated and their control plants occur in the third week. After that time point, the SPAD values tend to align with the values of their respective controls. This trend may be attributed to the higher growth rates observed in inoculated plants (Figure 2). The observed differences were, however, not statistically significant.

### 2.2. Debaryomyces hansenii Effect on Mineral Total Uptake

The total uptake of Cu, Fe, Mn, Zn, and P was assessed to provide a comprehensive understanding of the nutritional status of plants in the calcareous soil experiment at both 20 and 28 dpi.

At 20 dpi, an increase in the total uptake of all studied elements was evident in inoculated plants, both in sterilized and non-sterilized soil conditions. In the non-sterile soil, there were significantly higher uptakes for all elements in inoculated plants than in non-inoculated plants (Figure 5A,C,E,G,I).

The trend observed at 28 dpi closely mirrored that observed at 20 dpi, except for Cu in non-sterilized conditions (Figure 5H) and Fe in sterilized soil conditions, where no significant differences were observed (Figure 5D). However, there were significant differences among inoculated and non-inoculated plants in the uptake of the remaining elements (Zn, Mn, and P) in both sterilized and non-sterilized soil conditions (Figure 5B,F,J).

These results clearly confirm the ability of this yeast to promote nutrient absorption by the plant in calcareous soil and explain in part the growth promotion observed in those plants inoculated in comparison with control plants. This positions *D. hansenii* as a promising candidate for enhancing agriculture while aligning with the criteria outlined in the Green Deal.

In Table 1, Pearson’s correlation coefficients between determined elements are shown. This coefficient revealed an antagonism among P and the studied microelements, although it was only statistically significant for Cu (*p* ≤ 0.01, r = −0.4250), Fe (*p* ≤ 0.05, r = −0.3972), and Mn (*p* ≤ 0.05, r = −0.3807). Meanwhile, Fe showed a direct relationship with Cu (*p* ≤ 0.05, r = 0.3453) and Mn (*p* ≤ 0.05, r = 0.4102) and showed antagonism with Zn, apart from *p*, although non-significant (*p* = 0.7746). Finally, another significant interaction showed up between Cu and Zn (*p* ≤ 0.001, r = 0.5188).

### 2.3. Debaryomyces hansenii Effect on Gene Expression

As it has been previously explained in the introduction section, some rhizosphere microorganisms have the capability to induce ISR. To test whether *D. hansenii* is able to induce ISR, relative expression of several genes related to ET synthesis (*ACO1*) and ET transduction pathway (*EIN2* and *EIN3*), as well as *PAL*, related to SA biosynthesis responsible for SAR induction, was studied in leaves from cucumber plants grown in calcareous soil sterile or not, inoculated or not with *D. hansenii* one or two times over time. Samples were collected at 20 dpi and 28 dpi.

As shown in Figure 6, at 20 dpi, relative expression of ET-related genes *ACO1*, *EIN2,* and *EIN3* increased in inoculated plants grown in sterile soil (Figure 6A,C,E), while *PAL* only enhanced its relative expression level in plants grown in non-sterile soil (Figure 6G,H) in comparison with their respective uninoculated control plants. However, at 28 dpi, relative expression of all genes studied significantly increased in cucumber plants grown in non-sterile soil (Figure 6B,D,F,H). In the case of *EIN2* and *EIN3*, their relative expression levels were similar with single or double inoculation (Figure 6D,F). On the contrary, *ACO1* relative expression was negatively affected by the double inoculation (Figure 6B), while *PAL* was positively affected by the double inoculation (Figure 6H).

## 3. Discussion

Previous results from our group highlighted the ability of the yeast *D. hansenii* to induce Fe deficiency responses [2]. Nevertheless, its ability to compete in the rhizosphere with other coexisting microorganisms and to promote growth in these conditions remained unexplored until the culmination of this study. The validation of this yeast’s competence in the rhizosphere was a pivotal milestone towards its prospective application as an Fe biofertilizer, thus serving as the primary focus of this research.

Our results clearly confirm the ability of this yeast to promote plant growth by improving nutrient absorption by the plant in calcareous soil since all nutrients determined in the present work increased in inoculated plants compared with their respective uninoculated controls. Besides plant growth promotion, *D. hansenii* was also able to promote flower development, which would be traduced in a higher productivity of cucumber plants. This positions *D. hansenii* as a promising candidate for enhancing agriculture while aligning with the criteria outlined in the Green Deal.

Pearson’s correlation coefficient showed an antagonism among P, and the microelements Fe, Cu, and Mn which agrees with numerous references in the literature [51,69,70,71,72]. Sánchez-Rodríguez et al. [73,74] in studies conducted in calcareous soils with species with different sensitivities to Fe chlorosis, revealed that phosphate fertilization modifies the availability of Fe in the soil and aggravates Fe chlorosis in sensitive plants. On the other hand, P is found forming complexes with soil components, among these, Fe oxides notably contribute, serving as surfaces for phosphorus adsorption [75]. Coumarins, released in response to Fe deficiency, could potentially improve P availability for plants by releasing it from the Fe compounds with which it forms those iron oxides [51]. Interestingly, the regulation of *BGLU42* and *PDR9* expression, which codify key enzymes in coumarin biosynthesis, is mediated by FIT through MYB72, the convergence node between Fe deficiency responses and ISR. This approach harnesses the potential of beneficial microorganisms to improve the availability of both Fe and P without causing detrimental effects to either nutrient. This allows that, in spite of the inverse correlation existing between P, Fe, and Mn, the inoculation with *D. hansenii* significantly improved the acquisition of all these elements in both sterile and non-sterile soil. This improvement underscores the direct influence of this yeast on nutrient uptake, indicating its significant role in enhancing the absorption of these essential elements.

Besides Fe acquisition-related genes [2], *D. hansenii* induced the expression of several genes related to ET biosynthesis (*ACO1)*, ET signaling (*EIN2* and *EIN3*), and SA biosynthesis (*PAL*) in cucumber leaves from plants grown in calcareous soil (sterile or non-sterile) inoculated (single and double inoculated) with *D. hansenii* CBS767 at 20 and 28 dpi. These results suggest that *D. hansenii,* with other microorganisms, could induce both ISR and SAR. Our results agree with the previous ones recently published by García-Espinoza et al. [10], where the inoculation of cucumber and melon plants with the entomopathogenic fungus (EPF) *Metarhizium brunneum* Petch strain EAMa 01/58-Su induced the expression of several genes related to ET synthesis and transduction pathway, as well as *PAL* and *LOX* genes related to SA and JA synthesis. This relative expression enhancement was correlated with lethal and sublethal effects on *Spodoptera littoralis.* In this study, the authors showed that the ISR-SAR induction was not necessarily related to endophytic colonization, despite having important effects on insect pest fitness [10]. In a very recent work, it was shown that the ethylene synthesis enhancement induced by Fe deficiency confers resistance to *Botrytis cinerea* in *A. thaliana* plants [76]. In this way, *D. hansenii* could provide plants with a “state of readiness”, known as priming that could protect them against pathogens and/or insect pests. This priming has been described in response to rhizosphere microbes, EPF, or pathogens [26,77,78,79,80]. However, this is the first time that this state of readiness is described in response to yeast.

The enhancement of ET-related gene expression (Figure 6) could partially explain the higher number of flowers developed in inoculated plants with *D. hansenii,* since ET has been associated with the regulation of this process [81,82]. Although the mode of action of ET in the regulation of flower development is not totally understood yet, its involvement in flower development is totally accepted. However, the reports on ethylene’s role in the flowering process are controversial [82]. Several studies suggest that ethylene delays flower development in some species, such as Arabidopsis (*Arabidopsis thaliana*) and morning glory (*Pharbitis nil*). Conversely, the exogenous ET treatment has been found to promote the transition to flowering in pineapple (*Ananas comosus*) and urn plants (*Aechmea fasciata*) [83]. In the case of cucumber, it has been demonstrated that ET promotes female flower development [81], which aligns with our observations, although we evaluated the total number of flowers without taking into account their sex.

In conjunction, the results shown in the present work evidence the great ability of *D. hansenii* to improve nutrient acquisition and plant growth in Fe-deficient environments. Besides this, *D. hansenii* has also demonstrated its capability to induce systemic responses despite not being an endophytic microorganism. Although further studies are still necessary, our results would indicate that this yeast could induce ISR and SAR. However, as this aspect is explored further and elucidated, it is crucial to consider that the inclusion of *D. hansenii* as a viable candidate for use as a biofertilizer hinges predominantly on its demonstrated ability to compete effectively in the rhizosphere, as highlighted by our experiments.

## 4. Materials and Methods

### 4.1. Yeast Strain and Inoculum Preparation

The yeast strain *Debaryomyces hansenii* CBS767 (wild-type, Netherland collection; [84]) was prepared in liquid yeast peptone dextrose (YPD). This medium was composed of 1% yeast extract, 2% peptone, and 2% glucose by adding a precultured cell suspension. The preculture was generated by transferring colony-forming units (CFUs) from Petri dishes into YPD-containing flasks and then incubating them for 72 h at 150 rpm and 26 °C in the absence of light until they reached the exponential phase (2 d).

The final inoculum source of CBS767 was obtained by adding the preculture suspension to new YPD-containing flasks and incubating them under the same conditions. Following incubation, the inoculum was centrifuged at 6000 rpm for 4 min at 4 °C. The resulting pellet was resuspended in sterile distilled water (SDW), and it was adjusted to a final concentration of 10^8^ cells g^−1^ of soil.

### 4.2. Calcareous Soil Bioassay

#### 4.2.1. Plant Culture, Inoculation, Experimental Design, and Sampling

Commercial cucumber (*Cucumis sativus* L. cv Ashley) seeds were obtained (Rocalba S.A.; Girona, Spain) and sterilized by immersing them in a 10% (*v*/*v*) sodium hypochlorite solution for 10 min, rinsing with sterile deionized water, and then germinating in the dark within papers moistened with 5 mM CaCl_2_. After 2–3 d, the seedlings were transferred to a plastic mesh held over a half-strength nutrient solution and kept in the dark for 2 d. After that, cucumber seedlings were transferred to a hydroponic system with Romheld and Marschner nutrient solution [85] continuously aerated with 20 μM Fe-EDDHA (Ethylenediamine di-2-hydroxyphenyl acetate ferric, an Fe chelating agent), in which they remained for one week. Seedlings were grown in a growth chamber at 22 °C day/20 °C night, with relative humidity between 60–70% and a 14 h photoperiod at a photosynthetic irradiance of 300 μMol·m^−2^ s^−1^ provided by white, fluorescent light (10,000 lux). Finally, plants were individually transplanted into pots with 500 g of either sterile or non-sterile calcareous soil. The soil used to fill the pots was obtained from Santa Cruz (Córdoba; 37°47′03″ N 4°36′35″ W) and sterilized at 121 °C for 50 min twice. The physical-chemical properties and phosphorus and iron availability for the plant in the sampled soil are shown in (Table 2).

Plants were inoculated once or twice with CBS767 by irrigating with 100 mL of adjusted cell suspension to reach a final concentration of 10^8^ cells g^−1^ of soil: (i) At transplant and (ii) 20 d after transplant. Single and double-inoculated plants were considered, receiving one or both CBS767 applications, respectively. Control plants were irrigated with 100 mL of sterile deionized water at the same two time points.

Pots were organized in a fully randomized design with 10 replicates for each yeast treatment (single-inoculated, double-inoculated, or control) and soil condition (sterile or non-sterile; 3 yeast treatments × 2 soil combinations × 10 seedlings = 60 pots). Plants were maintained in a growth chamber under the same controlled conditions previously described over the experimental period (28 d). Plants were irrigated three times per week, adding the same volume of water per pot (100 mL).

#### 4.2.2. Assessments and Data Analysis

Fresh and dry weight, number of flowers, plant’s aerial part total mineral uptake of copper (Cu), iron (Fe), manganese (Mn), zinc (Zn), and phosphorus (P), and gene relative expression were determined at 20 and 28 d post-first inoculation (dpi), coinciding with the end of the experiment.

##### Mineral Total Uptake (MTU)

At each sampling time (20 and 28 dpi), the aerial part of plants was excised and oven-dried for 72 h at 70 °C (J. P. Selecta 140B, Barcelona, Spain). The resulting dry weight (dw) was then recorded, and samples were homogenized to a fine powder using an electric grinder (YellowLine A10; IKA-Werke, Staufen, Germany).

Mineralization was performed over 0.3 g of each sample, previously weighed in ceramic vessels, by a wet digestion process with a nitric and perchloric acid mixture [86]. Mineral concentrations were determined by Flame Atomic Absorption Spectroscopy (FAAS) for Cu, Fe, Mn, and Zn in a Perkin Elmer AAnalyst 200 spectrophotometer (Waltham, MA, USA), while P was determined in a BioTek PowerWave HT microplate reader spectrophotometer (Winooski, VT, USA) by the Molybdenum Blue method [87].

Finally, the mineral total uptake (MTU) was estimated as the total amount of each element expressed in µg (Cu, Fe, Mn and Zn) or mg (P) by the following equation:MTU=dw·mc
where “dw” is the dry weight (kg) of the plant sampled and “mc” is the mineral concentration (mg kg^−1^) of each measured element.

Performance characteristics such as the accuracy and linearity of the method were tested to ensure the analytical quality of these analyses. The accuracy was verified by using Certified Reference Materials (CRM; BCR-679, Community Bureau of Reference) of vegetal origin (white cabbage), analyzed in the same way as the samples. Furthermore, linearity was established through the calibration curves and the respective regression coefficient (*R*^2^) for each of the studied mineral elements. Finally, the limit of detection (LOD) and the limit of quantification (LOQ) were calculated as three and ten times the standard deviation of the blank measurements, respectively, in accordance with the UNE-EN 13804 [88] standards.

##### Measurement of Relative Chlorophyll Contents (SPAD Values)

The chlorophyll content in leaves was assessed on the central region of each leaf using the Minolta chlorophyll meter SPAD-502. Three independent SPAD measurements were carried out in totally expanded apical leaves from each plant of the different treatments (10 plants per treatment). Data represent the mean of 10 independent biological replicates ± SE.

##### qRT-PCR Analysis

PCR analysis was executed in accordance with García et al. [89]. Total RNA extraction was carried out using the Tri reagent solution (Molecular Research Center, Inc., Cincinnati, OH, USA). Subsequently, cDNA synthesis was performed using M-MLV reverse transcriptase (Promega, Madison, WI, USA), with 3 μg of DNase-treated leaf RNA as the template and random hexamers as primers.

Gene expression analysis by qRT-PCR was performed using a Bio-Rad CFX connect thermal cycler (Hercules, CA, USA). The amplification profile consisted of an initial denaturation and polymerase activation step at 95 °C for 3 min, followed by 40 cycles of amplification and quantification (90 °C for 10 s, 57 °C for 15 s, and 72 °C for 30 s). Additionally, a final melting curve stage ranging from 65 to 95 °C with a 0.5 °C increment over 5 s was conducted to verify the absence of primer dimers or non-specific amplification products.

PCR reactions were set up in 20 μL of SYBR Green Bio-Rad PCR Master Mix, following the manufacturer’s instructions. Controls containing water instead of cDNA were included to check for contamination in the reaction components. Standard dilution curves were performed for each primer pair to confirm the appropriate efficiency of amplification (E = 100 ± 10%). *ACO1*, *EIN2*, *EIN3*, and *PAL* were amplified by using the primers previously described by García-Espinoza et al. [10]. Constitutively expressed *CYCLO* and *ACTIN* genes were used as reference genes to normalize qRT-PCR results. The relative expression levels were calculated from the threshold cycles (Ct) values and the primer efficiencies by the Pfaffl method [90]. Each PCR analysis was conducted on three biological replicates (each biological replicate was a mixture of the roots of two plants), and each PCR reaction was repeated twice.

##### Statistical Analysis

Data underwent normality and homogeneity of variances tests to confirm their appropriateness for statistical analysis. For data meeting both normality and homogeneity of variances assumptions, a one-way ANOVA was conducted for each soil (sterile or non-sterile) with plant growth, MTU, and relative gene expression levels as dependent variables and treatments (control, single inoculation, and double inoculation) as independent variables. When the assumptions of normality and homogeneity of variances were not met, a non-parametric Kruskal–Wallis one-way AOV was conducted. The comparison of the means was performed using Fisher’s Protected Least Significant Differences (LSD) and Dunn’s test, respectively, at *p* ≤ 0.05, *p* ≤ 0.01, or *p* ≤ 0.001 from ANOVA or Kruskal–Wallis one-way AOV, respectively. Additionally, the associations between different mineral element concentrations were estimated through Pearson correlation coefficients (r; Table 1). Data were analyzed using the STATISTIX software v.10 (Analytical Software, Tallahassee, FL, USA).

## Figures and Tables

**Figure 1 ijms-25-05729-f001:**
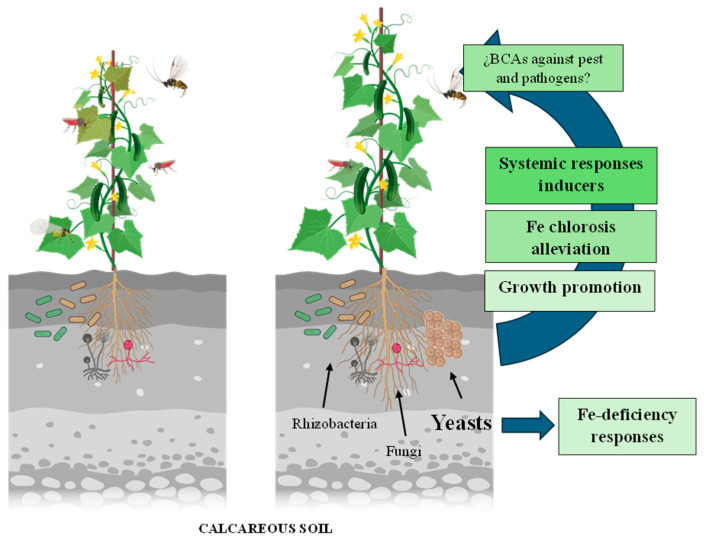
Schematic representation of the microbiome (rhizobacteria, rhizofungi, and/or yeast) role in activating local and systemic responses to biotic and abiotic stresses. Rhizosphere microorganisms trigger induced resistance through long-distance signals transported by the vascular system. Additionally, airborne signals can systemically amplify plant defense mechanisms against pathogen infections or infestations by herbivores in induced plant tissues, leading to notably reduced damage. Locally, some rhizosphere microorganisms can induce Fe deficiency responses, contributing to an improvement of plant nutrition and, as consequence, promoting plant growth and alleviating Fe chlorosis [1,2,10,11,26]. Created with BioRender.com.

**Figure 2 ijms-25-05729-f002:**
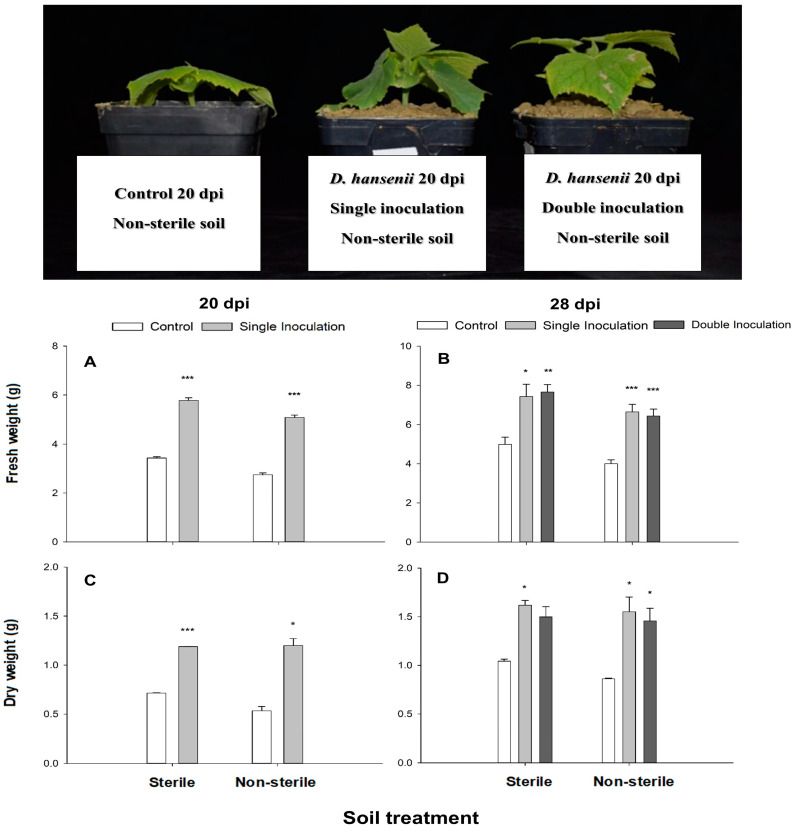
Effect of the inoculation with *D. hansenii* CBS767 on the growth of cucumber plants grown in calcareous soil at 20 dpi (upper part of the figure). Fresh (**A**,**B**) and dry weight (**C**,**D**) of cucumber plants grown in calcareous soil (sterile or non-sterile), non-inoculated (control), or inoculated (single and double inoculated) with *D. hansenii* CBS767 at 20 dpi (**A**,**C**) and 28 dpi (**B**,**D**). Values represent mean ± SE of 4 different replicates (n = 4). *, ** or *** asterisks show significant statistical differences between inoculated treatments and control at *p* ≤ 0.05, *p* ≤ 0.01, and *p* ≤ 0.001, respectively, according to Fisher’s LSD test.

**Figure 3 ijms-25-05729-f003:**
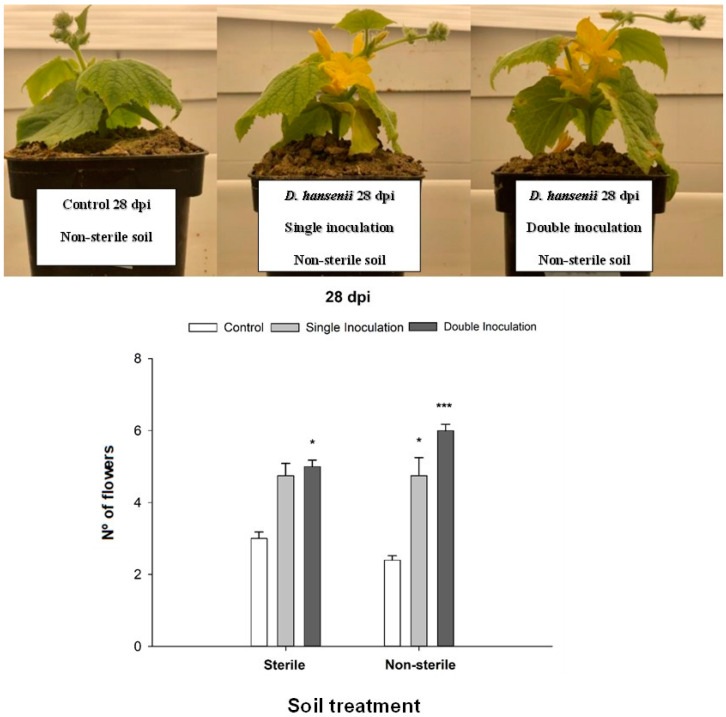
Effect of the inoculation with *D. hansenii* CBS767 on the number of flowers in cucumber plants grown in calcareous soil at 28 dpi (upper part of the figure). Number of flowers in cucumber plants grown in calcareous soil (sterile or non-sterile), non-inoculated (control), or inoculated (single and double inoculated) with *D. hansenii* CBS767 at 28 dpi (lower part of the figure). Values represent mean ± SE of 4 different replicates (n = 4). * or *** asterisks show significant statistical differences between inoculated treatments and control at *p* ≤ 0.05 and *p* ≤ 0.001, respectively, according to Fisher’s LSD test.

**Figure 4 ijms-25-05729-f004:**
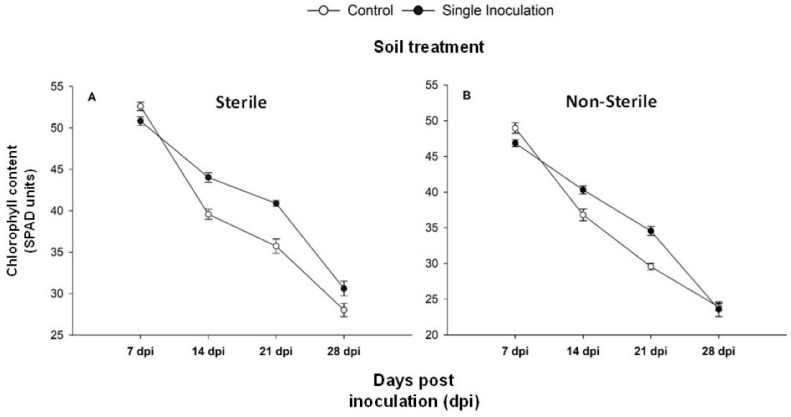
Chlorophyll content (SPAD units) of cucumber plants grown on calcareous soil, sterile (**A**) or non-sterile (**B**) non-inoculated (control) or inoculated (single inoculated) with CBS767 at 7, 14, 21, and 28 dpi. Values represent mean ± SE of 4 different replicates (n = 4).

**Figure 5 ijms-25-05729-f005:**
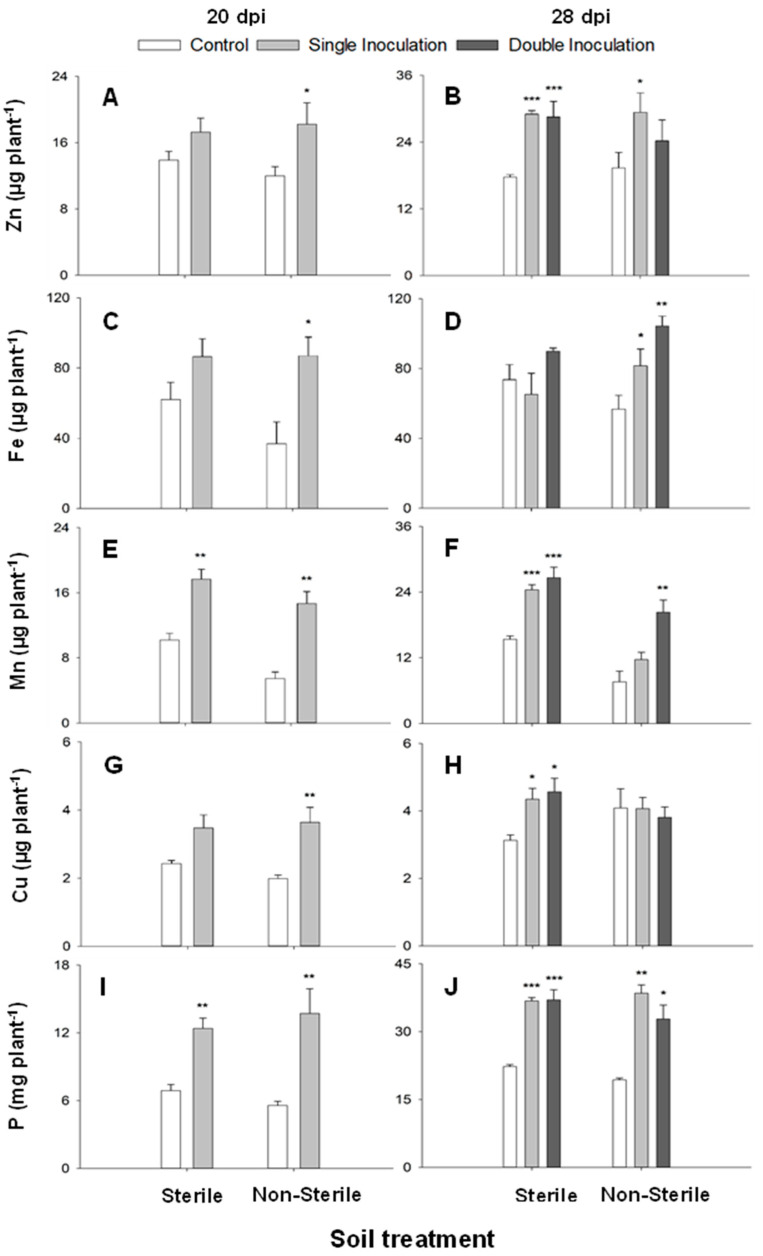
Total mineral uptake (µg plant^−1^) of Zn (**A**,**B**), Fe (**C**,**D**), Mn (**E**,**F**), Cu (**G**,**H**), and P (mg plant^−1^; (**I**,**J**)) of cucumber plants grown in calcareous soil (sterile or non-sterile), non-inoculated (control), or inoculated (single and double inoculated) with *D. hansenii* CBS767 at 20 dpi (**A**,**C**,**E**,**G**,**I**) and 28 dpi (**B**,**D**,**F**,**H**,**J**). Values represent means ± SE of 4 different replicates (n = 4). *, ** or *** asterisks show significant statistical differences between inoculated treatments and control at *p* ≤ 0.05, *p* ≤ 0.01, and *p* ≤ 0.001, respectively, according to Fisher’s LSD test.

**Figure 6 ijms-25-05729-f006:**
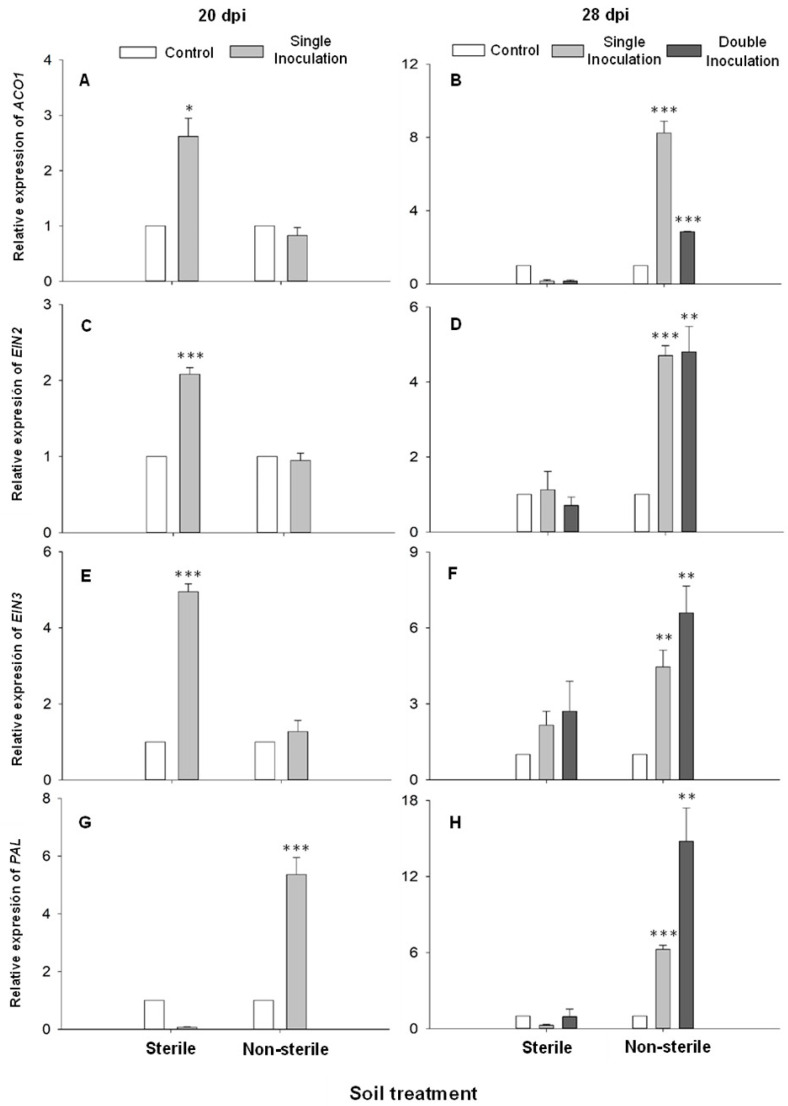
Relative expression of ET biosynthesis (*ACO1*; (**A**,**B**)), ET transduction (*EIN2* and *EIN3*; (**C**–**F**)), and SA biosynthesis (*PAL*; (**G**,**H**)) genes in cucumber leaves from plants grown in calcareous soil (sterile or non-sterile), non-inoculated (control), or inoculated (single and double inoculated) with *D. hansenii* CBS767 (10^6^ cells mL^−1^), at 20 dpi (**A**,**C**,**E**,**G**) and 28 dpi (**B**,**D**,**F**,**H**). Values represent means ± SE of 4 different replicates (n = 4). *, ** or *** asterisks show significant statistical differences between inoculated treatments and control at *p* ≤ 0.05, *p* ≤ 0.01, and *p* ≤ 0.001, respectively, according to Fisher’s LSD test.

**Table 1 ijms-25-05729-t001:** Pearson’s correlation coefficients between the mineral concentration (mg kg^−1^ dw) of the different elements in cucumber plants grown on calcareous soil.

Mineral Concentration (mg kg^−1^ dw)	Mineral Concentration (mg kg^−1^ dw)	Association Coefficient Value (*r*)	*p*-Value
Zn	Fe	−0.0480	0.7746
	Mn	−0.2779	0.0912
	Cu	0.5188	≤0.001
	P	−0.1269	0.4476
Fe	Zn	−0.0480	0.7746
	Mn	0.4102	≤0.05
	Cu	0.3453	≤0.05
	P	−0.3972	≤0.05
Mn	Zn	−0.2779	0.0912
	Fe	0.4102	≤0.05
	Cu	−0.1287	0.4411
	P	−0.3807	≤0.05
Cu	Zn	0.5188	≤0.001
	Fe	0.3453	≤0.05
	Mn	−0.1287	0.4411
	P	−0.4250	≤0.01
P	Zn	−0.1269	0.4476
	Fe	−0.3972	≤0.05
	Mn	−0.3807	≤0.05
	Cu	−0.4250	≤0.01

**Table 2 ijms-25-05729-t002:** Physical-chemical properties and phosphorus and iron availability for the plant (average) in the calcareous soil used.

Clay g kg^−1^	Organic Carbon g kg^−1^	CaCO_3_ g kg^−1^	pH_1:2.5_	EC_1:5_ dS m^−1^	CEC cmol kg^−1^	P_Olsen_ mg kg^−1^	Fe_DTPA_ mg kg^−1^
370	9.3	338	7.9	1.50	31.3	13.4	4.3

CaCO_3_: Carbonates content. pH_1:2.5_: Soil pH in the extract 1:2.5 (soil: deionized water). EC_1:5_: Electrical conductivity in the extract 1:5 (soil: deionized water). CEC: Cation exchange capacity. P_Olsen_: P available on soil. Fe_DTPA_: Labile Fe in the soil.

## Data Availability

Data is contained within the article.

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
