# Peer review of "Exploring the Role of Debaryomyces hansenii as Biofertilizer in Iron-Deficient Environments to Enhance Plant Nutrition and Crop Production Sustainability"

_ijms, 2024, doi:10.3390/ijms25115729_

Round 1
Reviewer 1 Report
Comments and Suggestions for Authors
Thanks for this valuable research, although some points must be improved, such as the abstract, which must have sentences about the material and methods used in this research.
Some abbreviations need correction, such as in line 109 lacks consistency between the full name and its abbreviation, so check all the abbreviations throughout the research.
The results are presented in paragraph before the results section
Figure 3 lacks A, B, C, and D, despite their presence in the description below.
Table 1 first column, is overlapped
Comments on the Quality of English LanguageMinor editing of English language is required
Author Response
First and foremost, we are delighted to receive the decision of acceptance (with minor review) for our manuscript titled: " Exploring the role of Debaryomyces hansenii as biofertilizer in iron-deficient environments to enhance plant nutrition and crop production sustainability." In response, we are submitting the revised version of the manuscript along with a detailed response to the review letter. We have addressed each reviewer's comment and recommendation individually, resulting in a significant enhancement of the paper's overall structure and comprehensibility.

Reviewer 2 Report
Comments and Suggestions for Authors
The work is interesting, although in my opinion more research could be done, corresponding to the topic of the work.I kindly ask you to respond to the following points:
1. Please write an explanation of the abbreviation FeEDDHA so that it can be understood by other readers
2. Please correct Table 1 as it is unreadable
3. in which oven the samples were dried (4.2.2.1)
4. There is no summary or conclusion, please complete this.
5. When testing for chlorophyll content, was the maximum depth to which the sample can be
inserted into the measuring head slot, thus maintaining a constant measuring point.
After taking into account the above comments, the publication can be published in the journal IJMS .
Author Response

(The authors gave the same response as above.)
